# Associations of Biopterins and ADMA with Vascular Function in Peripheral Microcirculation from Patients with Chronic Kidney Disease

**DOI:** 10.3390/ijms24065582

**Published:** 2023-03-15

**Authors:** Samsul Arefin, Lars Löfgren, Peter Stenvinkel, Anna B. Granqvist, Karolina Kublickiene

**Affiliations:** 1Department of Clinical Science, Intervention & Technology, Division of Renal Medicine, Karolinska Institutet, 141 86 Stockholm, Sweden; samsul.arefin@ki.se (S.A.);; 2Early CVRM, BioPharmaceuticals R&D, AstraZeneca, 431 50 Mölndal, Sweden

**Keywords:** chronic kidney disease, biopterins, asymmetric dimethylarginine, amino acids, vascular function

## Abstract

We hypothesized that patients with chronic kidney disease (CKD) display an altered plasma amino acid (AA) metabolomic profile that could contribute to abnormal vascular maintenance of peripheral circulation in uremia. The relationships between plasma AAs and endothelial and vascular smooth muscle function in the microcirculation of CKD patients are not well understood. The objective of this study is to investigate to what extent the levels of AAs and its metabolites are changed in CKD patients and to test their relationship with endothelial and vascular smooth muscle function. Patients with CKD stages 3 and 5 and non-CKD controls are included in this study. We report that there was a significant reduction of the biopterin (BH_4_/BH_2_) ratio, which was accompanied by increased plasma levels of BH_2_, asymmetric dimethylarginine (ADMA) and citrulline in patients with CKD-5 vs. CKD-3 vs. controls. In vivo augmentation index measurement showed a positive association with ADMA in all participants. The contribution of nitric oxide, assessed by ex vivo assay, showed a negative association with creatinine, ADMA and citrulline in all participants. In CKD-5, BH_4_ negatively correlated with ADMA and ornithine levels, and the ex vivo endothelium-mediated dilatation positively correlated with phenylalanine levels. In conclusion, uremia is associated with alterations in AA metabolism that may affect endothelium-dependent dilatation and vascular stiffness in microcirculation. Interventional strategies aiming to normalize the AA metabolism could be of interest as treatment options.

## 1. Introduction

Chronic kidney disease (CKD) has emerged as a global health complication with a prevalence of 8–16%, which ultimately leads to end-stage kidney disease (ESKD) [1]. CKD is also an important hazard for cardiovascular disease (CVD) mortality and morbidity, and CVD complications increase with the extent of renal impairment. The underlying mechanisms are far from clear, but as for vascular disease, the interplay between the endothelium and the vascular smooth muscle cell (VSMC) layer is affected due to interrelated mechanisms, contributing to the exaggeration of athero- and arteriosclerosis simultaneously [2].

The endothelium is a metabolically active organ essential for preserving vascular equilibrium, and its contribution is mediated through endothelium-derived factors. Nitric oxide (NO) is one of the major players in endothelium-dependent relaxation and the maintenance of optimal cardiovascular health. Tetrahydrobiopterin (BH_4_) plays a crucial role as a cofactor for all three isoforms of nitric oxide synthase (NOS; endothelial NOS (eNOS), neuronal NOS (nNOS) and inducible NOS (iNOS)) [3], along with other amino acids (AAs), e.g., L-arginine, L-phenylalanine, citrulline or ornithine to generate NO (Figure 1). When the bioavailability of BH_4_ is restricted, it favors NOS uncoupling, with the facilitation of superoxide (O_2_^−^) production at the expense of NO, further amplifying the effects of oxidative stress. Reduced NO bioavailability is a common feature of CKD, including in patients with ESKD [4]. Several processes affecting the production and activity of eNOS contribute to low NO bioavailability in CKD [4]. The oxidation of BH_4_, primarily by peroxynitrite (ONOO^−^) forming 7,8-dihydrobiopterin (BH_2_), which promotes eNOS uncoupling, may have a negative impact on BH_4_ levels and NO bioavailability (Figure 1) [5].

An impaired endothelial function, a key initiating step in the pathogenesis of CVD, has been reported in CKD [4,6]. In addition, when considering altered microcirculation in patients undergoing peritoneal dialysis, an impairment of NO contribution has been proposed [7]. Asymmetric dimethylarginine (ADMA), a strong and independent risk factor for CKD, is an endogenous inhibitor of NOS that attenuates NO production and may enhance reactive oxidative species (ROS) generation [8]. As the BH_4_:BH_2_ ratio is important for eNOS coupling and uncoupling, strategies increasing BH_4_ levels have been suggested to improve vascular hemodynamics [9]. Microvascular disease in CKD is the base of the iceberg in cardiovascular comorbidity; however, most of the CKD-related CVD research has focused on macrovascular consequences. Therefore, there is a serious need for intensive research in the field of microcirculation in CKD, specifically in humans.

Recent studies have demonstrated that AA contributions are related to the regulation of endothelial cell proliferation, migration, survival and even function, thus further emphasizing that AAs play an important role in maintaining vascular homeostasis [10,11]. One of the important AAs is L-arginine, which is synthesized from glutamic acid and catalyzed by a number of enzymes. L-arginine in the body is either broken down by arginase into urea and ornithine or by NOS into equal amounts of citrulline and NO (Figure 1). Citrulline can be converted to L-argininosuccinate and subsequently to L-arginine again, thus also contributing to the elevation of the plasma and tissue levels of L-arginine. The other AA involved in the regulation of endothelial function is L-phenylalanine, which may promote BH_4_ synthesis and increase nitrite levels by activating the guanosine triphosphate cyclase hydrolase-1 (GCH1) pathway, thereby attenuating ROS and increasing NO levels [12]. BH_4_ is synthesized from GTP in a reaction where GCH1 mediates the rate-limiting step. The modulation of GCH1 expression controls BH_4_, NO and cardiovascular function [13]. L-phenylalanine controls GCH1 via feed-forward regulation and via an allosteric protein communication with the GCH1 feedback regulatory protein (GFRP) (Figure 1) [12]. Since oral administration with L-phenylalanine causes several folds of increase in plasma biopterin concentrations, the GCH1–GFRP complex has been suggested to be functional in humans [14]. In addition, proline is an important constituent of collagen; it accounts for up to 10% of all AAs in collagen [15] and regulates dehydration stress, redox, cell proliferation, differentiation and death, among other processes [16].

While the role of altered AA metabolism in the pathogenesis of CKD has been studied previously, the existing, if any, relationships between plasma AAs and endothelial and VSMC function in CKD patients are not well understood. Additionally, the extent to which CKD-associated AA metabolism contributes to the abnormal vascular maintenance of peripheral circulation remains unclear as the majority of the studies were performed either in animals or in bigger arteries. Therefore, there is a gap in understanding the precise relationship between AA metabolism and vascular dysfunction, particularly in CKD patients. The results of our study may provide more insights into changes in plasma AA levels and metabolites in CKD patients and determine their relationship with endothelial and VSMC function in microcirculation.

Thus, we hypothesized that the severity of renal disease would affect plasma AA metabolism, as reflected by levels in circulation, and potentially associate with changes in vascular function and structure. We, therefore, aimed to measure levels of AAs and their metabolites and investigate the relationship between BH_4_, BH_2_, ADMA and other AAs and vascular parameters by applying in vivo and ex vivo techniques to assess function and structure in peripheral microcirculation from non-CKD controls and CKD-3 and CKD-5 patients. In vivo vascular function was assessed by EndoPAT, previously reported to correlate with endothelial function in the coronary circulation and predictive for CVD [17,18], while isolated small arteries from peripheral circulation were used to assess ex vivo vascular function and structure [7].

## 2. Results

### 2.1. Study Population

The age, demographic, biochemical, and clinical characteristics of all participants are shown in Table 1. The biochemical parameters showed a typical profile of CKD-5/CKD-3 patients, with significantly higher levels of creatinine, triglycerides and phosphate and a gradual decline of the glomerular filtration rate (eGFR) and lower albumin levels compared to controls. Both systolic blood pressure (SBP) and diastolic blood pressure (DBP) were increased in CKD-5 patients, while in CKD-3 patients, only SBP was increased. The age and BMI were matched between control and CKD-5/or CKD-3 patients. The percentage of males was relatively higher in CKD-5 (72%) and CKD-3 (75%) patients compared to the control (26%) group.

### 2.2. Plasma Amino Acids and Their Metabolites

Plasma AA metabolites were measured to determine whether patients with CKD have altered plasma AA metabolomic profiles and their potential role in the abnormal vascular maintenance of peripheral circulation in uremia and their relationships with endothelial and VSMC function in CKD patients. A significant reduction in the BH_4_/BH_2_ ratio was observed in patients with CKD-5 compared with the non-CKD control and CKD-3 participants (control 1.40 nM/nM (1.10–2.0) vs. CKD-3 1.55 nM/nM (1.10–2.32) vs. CKD-5 0.50 nM/nM (0.25–1.0); Figure 2A), which suggests a decreased conversion of BH_2_ to BH_4_ or reduced activity of dihydrofolate reductase. The plasma levels of BH_2_ (control 2.50 nM (2.0–3.1) vs. CKD-3 2.10 nM (1.85–3.45) vs. CKD-5 6.10 nM (4.40–7.60)) and ADMA (control 0.50 µM (0.50–0.50) vs. CKD-3 0.50 µM (0.50–0.60) vs. CKD-5 0.70 µM (0.60–0.75)) were increased with the progression of CKD (Figure 2C,D). In contrast to BH_2_, plasma levels of BH_4_ were similar between the three groups studied (control 4.4 nM (2.87–7.12) vs. CKD-3 3.65 nM (2.80–9.15) vs. CKD-5 2.90 nM (1.65–5.30); Figure 2B).

Plasma AAs were measured to investigate whether the severity of CKD altered their profiles. With the progression of CKD, the levels of citrulline [control 23.95 µM (20.65–26.95) vs. CKD-3 37.50 µM (33.55–46.75) vs. CKD-5 63.40 µM (56.70–81.71)] and proline (control 89.25 µM (76.55–113.8) vs. CKD-3 131.5 µM (108.3–171.3) vs. CKD-5 147.0 µM (130.5–198.0)) were increased both in CKD-3 and CKD-5 compared to the control group (Figure 3B,E). We did not observe any difference in arginine, phenylalanine and ornithine levels between the three groups (Figure 3A,C,D).

Plasma AA ratios were calculated to evaluate the kidneys’ ability for AA interconversions and the presence of urea cycle dysfunction. Patients from different CKD stages showed variations in the AA ratio along the course of the disease. The ratio of Arg/Cit (Figure 4A) was significantly reduced, whereas the ratio of Cit/Orn (Figure 4B) and Pro/Orn (Figure 4C) increased with the advancement of CKD.

### 2.3. Correlations between In Vivo and Ex Vivo Measurements and AA Metabolites

Correlation analysis was used to establish the relationships between plasma AAs and endothelial and VSMC function and structure in CKD patients. The BH_4_/BH_2_ ratio showed a positive correlation with eGFR and a negative correlation with creatinine; conversely, ADMA showed a negative correlation with eGFR and a positive correlation with creatinine in all participants (Figure 5A–D). The BH_2_ ratio negatively correlated with eGFR (Figure 5E), further strengthening the findings of the increased BH_2_ with the progression of CKD, as presented in Figure 2. In vivo measurement of peripheral vascular endothelial function, reflected in the reactive hyperemia index (RHI), did not show any associations with AA metabolites. However, stiffness, reflected in the augmentation index (AI@75), showed a positive association with soluble ADMA and a negative association with eGFR in all participants (Figure 5F,G and Appendix A). An ex vivo functional study was performed to investigate endothelial and VSMC function and to calculate the contribution of endothelial mediators NO and EDHF. The contribution of NO to the endothelium-dependent dilatation in the ex vivo experiment showed a negative association with creatinine, ADMA and citrulline (Figure 5H–J), and the contribution of the endothelium-derived hyperpolarizing factor (EDHF) positively correlated with ADMA levels (Figure 5K) in all participants. Of note, the ex vivo experiments were performed only in the CKD-5 and non-CKD control groups.

The contribution of the percentage of NO and EDHF was calculated using pharmacological tools (Appendix A). As NOS and COX inhibition entirely blocks NO and PGI2, if any, the remaining response is considered EDHF-mediated dilatation. We have observed that the inhibition of only COX does not change the dilation (Appendix A), suggesting that the PgI2 response is absent in those resistance arteries, and we have considered that only NO and EDHF are present in those arteries. In CKD-5 patients, endothelium-mediated dilatation was positively correlated with phenylalanine (Figure 6A), and BH_4_ was negatively correlated with ADMA and ornithine, respectively (Figure 6B,C).

## 3. Discussion

In the present study, the circulating AAs, their metabolites, and their association with vascular function and structure were investigated in patients with CKD at different stages, reflecting the deterioration of kidney function. The key findings were: (a) the plasma AAs and their metabolite profile alterations were more pronounced in advanced renal disease, (b) the BH_4_/BH_2_ ratio gradually decreased with an increase in creatinine levels and disease progression, (c) CKD-5 showed a significant increase in plasma BH_2_, ADMA and citrulline compared to non-CKD controls, (d) ADMA was positively associated with vascular stiffness and negatively associated with NO contribution and (e) in CKD-5 patients, phenylalanine was positively associated with endothelium-dependent dilatation. Overall, the findings suggest a complex interplay between AA metabolites and the mediators of endothelial function and structure that would adversely affect vascular function in uremia and most likely enhance the risk for CVD.

The BH_4_/BH_2_ plasma ratio decreased significantly while the levels of BH_2_ increased with the progression of CKD. When the demand for BH_4_ increases in healthy conditions, BH_4_ is formed from guanosine triphosphate (GTP) [19] or is produced through a reduction in oxidized BH_2_. Under oxidative stress, a common feature of CKD [20], BH_4_ acts as a radical scavenger and can be oxidized to BH_2_. The reduced BH_4_/BH_2_ ratio suggests a decreased conversion of BH_2_ to BH_4_ or the reduced activity of dihydrofolate reductase [21] in our CKD-5 patients. We did not see differences in BH_4_ levels between CKD patients at different disease stages. The data are in line with and extend the findings from Yokoyama, K et al., who demonstrated unchanged BH_4_ levels among CKD-5 patients versus healthy controls [22]. However, BH_2_ levels increased with the advancement of CKD, which could have an impact on biological activity due to its competitive binding with eNOS [23]. Therefore, although BH_4_ levels did not change in plasma due to compromised renal function, the reduced BH_4_/BH_2_ ratio and elevated BH_2_ could contribute to the decreased NO production in our CKD patients [23]. This concurs with our results from the ex vivo experiment, illustrating that the NO contributory pathway is impaired in uremic arteries vs. controls (S2). It is important to stress that other mechanisms might also play a role, e.g., several studies have suggested the importance of a pro-oxidative environment and its effect on NO bioavailability using experimental models of kidney failure [24,25]. Several cellular mechanisms can act as a source of oxidative stress among patients with CKD. The triad of oxidative stress, chronic inflammation and endothelial dysfunction, which results in the decreased bioavailability of NO, is acknowledged as a bidirectional vicious cycle that exacerbates the relationship between CKD and systemic complications [26]. It has been suggested that reduced mitochondrial function and an increased generation of mitochondrial ROS are potential factors leading to heightened oxidative stress in CKD [27], which, in turn, can impact the availability of NO.

We argue that ADMA may serve as a feasible candidate for the reduced NO contribution in patients with advanced CKD. This is supported by our observation of a gradual increase in ADMA levels correlating to the reduced GFR and the progression of CKD, data that are in line with previous reports [8,28]. Since ADMA acts on vascular function by inhibiting eNOS, it is feasible to suggest that higher ADMA levels would inhibit NO production, resulting in vasoconstriction, hypertension, immune dysfunction and ultimate endothelial dysfunction in CKD [29]. Additionally, ADMA-associated endothelial dysfunction and vascular remodeling could occur via the amplification of oxidative stress. The negative correlation between ADMA and NO contribution in CKD-5 patients supports this and suggests a link towards upstream vascular malfunction that contributes to the arteriosclerosis, cardiovascular events and deaths seen in CKD patients, as previously indicated [30].

In our study, changes in plasma concentrations of AAs (particularly of citrulline and proline) appeared already in CKD-3 and were higher in the advanced stage of the disease. The liver is primarily responsible for synthesizing citrulline via the urea cycle, and subsequently, renal cells break it down into arginine. In our study, increased citrulline concentration occurred when eGFR decreased below 45 mL/min. In addition, the relative presence of arginine compared with citrulline (Arg/Cit) was progressively lower with the advancement in kidney failure, suggesting a lower activity of argininosuccinate synthase and/or argininosuccinate lyase. Our finding of altered citrulline metabolism due to a decline in renal function concurs and further strengthens data from a study showing higher plasma citrulline in patients with CKD [31,32]. We did not observe any significant changes in plasma ornithine concentrations between the different groups. Ornithine serves as a transitional molecule in the urea cycle and undergoes conversion into citrulline. As CKD advances to stage 5, the rise in plasma citrulline levels (reflected by an increase in the Cit/Orn ratio) suggests an elevation in ornithine breakdown, leading to the production of citrulline and, potentially, proline synthesis. This is supported by the increased Pro/Orn ratio seen during the advancement of kidney disease. The findings indicate that the reduced GFR and urinary excretion lead to the accumulation of ornithine, which triggers the production of citrulline and proline.

The increased vascular stiffness, together with higher plasma ADMA levels in our study, deserves some attention. ADMA has been shown to be associated with left ventricular hypertrophy [33], high sympathetic activity [34] and increased cardiovascular risk [30,34]. Presently available information suggests that administering ADMA results in decreased blood flow in the forearm [35] while increasing systemic vascular resistance, intima-media thickness [36] and the augmentation index [37], further strengthening the suggestion that ADMA may serve as a risk factor in CKD. We have also investigated the possible association of circulating proline levels with in vivo and ex vivo stiffness measurements as proline is an important constituent of collagen and may affect stiffness and reduce elasticity [38]. Although the proline levels were increased with the progression of CKD, proline showed no association with stiffness or elasticity in our study. The reason behind this is unclear and warrants further research.

We have observed that phenylalanine is positively associated with endothelium-dependent dilatation in CKD-5 patients, which is in line with a recent study by Heikal et al. [39], in which the GCH1–GFRP complex was activated by phenylalanine, leading to the restoration of vascular function in spontaneously hypertensive rats. This suggests that higher phenylalanine levels could positively affect endothelium-dependent dilatation in CKD patients. Phenylalanine plays an important role in the synthesis of BH_4_. Although we did not find a direct association of BH_4_ with NO contribution or endothelial function, the positive association between phenylalanine and endothelial function might indirectly give support for the involvement of BH_4_ levels with endothelial function as phenylalanine favors BH_4_ production [39,40].

In summary, the BH_4_/BH_2_ ratio is decreased in CKD. The AA metabolism is impaired in CKD and elevated ADMA levels are associated with higher vascular stiffness and reduced NO contribution. Several studies have suggested the exogenous supplementation of AAs, e.g., arginine [41], citrulline [42] and L- phenylalanine [43], as a potential therapeutic option to target vascular disease; however current evidence is contradictory and may depend on differences in dose and administration route. In conclusion, we suggest that uremia is associated with alterations in AA metabolism that may affect vascular function in CKD towards an increased risk for CVD. Future research is warranted on the potential effects of AAs and their metabolites on vascular function and structure to support the potential of AA application as a therapeutic strategy.

## 4. Materials and Methods

### 4.1. Study Participants

The investigation was undertaken with the approval of the Ethical Committee ‘‘Regionala Etikprövningsnämnden i Stockholm’’, Sweden, with the informed consent of each patient and in accordance with the principles outlined in the Declaration of Helsinki. We have included 25 patients with CKD-5 (eGFR < 15 mL/min/1.73 m^2^) undergoing living donor renal transplantation, 16 patients with CKD-3 (eGFR 30–45 mL/min/1.73 m^2^) and 22 kidney donors who served as non-CKD controls (eGFR > 85 mL/min/1.73 m^2^). One participant from the CKD-3 group was excluded (eGFR < 30 mL/min/1.73 m^2^). For the in vivo experiments to assess vascular function and the measurement of AA metabolites from plasma, all three groups were included. However, for the ex vivo experiments, only the CKD-5 and control groups were included due to the availability of the fat biopsies. For the baseline characteristics, the concentrations of serum creatinine, albumin, calcium, phosphate, triglycerides, cholesterol, high-density lipoprotein (HDL) cholesterol, high-sensitivity C-reactive protein (hsCRP) and glycated hemoglobin A1c (HbA1c) were assessed at the Department of Laboratory Medicine, Karolinska University Hospital, Huddinge, Sweden.

### 4.2. Collection of Blood Samples for the Measurements of Amino Acids and Biopterines

Venous blood samples were obtained from CKD-3, CKD-5 and control participants. For the analysis of AAs, blood was collected in K3-EDTA tubes, and plasma was prepared within 30 min and stored at −80 °C. For the analysis of BH_2_ and BH_4_, the blood collection procedure by Fekkes and Voskuilen-Kooijman was applied [44]. Briefly, blood was collected in K3-EDTA tubes containing dithiothreitol (DTT) at a final concentration of 0.1% and stored for 2.5 h (±30 min) at room temperature in the dark before being centrifuged. Plasma was prepared and immediately stored at −80 °C. DTT was used to protect the samples from oxidation and the side-chain cleavage of BH_4_.

### 4.3. Analysis of Amino Acids in Plasma

For the analysis of the AAs proline, ornithine, phenylalanine, arginine, citrulline and ADMA, 200 µL internal standard solution (47.5% acetonitrile, 47.5% methanol and 5% water with 10 µM L-arginine 13C6 15N4, L-ornithine 13C 15N2, L-citrulline 13C5 D4, L- proline 13C5 15N and L-phenyl-D5-alanine-D3 and 0.1 µM ADMA-D7) was added to 25 µL ice-cold plasma in a polypropylene 96-well plate. Samples were mixed for 5 min and centrifuged for 10 min at 4000 rpm at 4 °C; 150 µL of the sample extract was transferred to a new deep-well plate ready for analysis by liquid chromatography–mass spectrometry (LC-MS). Chromatographic separation was achieved by hydrophilic interaction chromatography (HILIC) using a 5 cm × 2.1 mm BEH amide column (Waters) in a gradient run, where mobile phase A was 95% acetonitrile with 5 mM ammonium formate (pH3) and mobile phase B was 10 mM ammonium formate (pH3) in water. The following gradient was applied for the AAs: 0% B for 0–0.5 min, then from 0–55% B from 0.5–3 min, 55% kept to 3.5 min, followed by equilibration from 3.5–5 min at 0% B prior to the next sample injection. Two µL sample extract was injected. The sample plates were kept at 10 °C, and the column was held at 30 °C with a flow rate of 0.4 mL/min.

The analyte was quantified in positive electrospray mode using multiple reaction monitoring (MRM) with the following MRM transitions, cone voltages and collision energies: L-arginine (175 > 70, 20/20), L-arginine 13C6 15N4 (185 > 75, 20/20), L-ornithine (133 > 70, 15/15), L-ornithine 13C 15N2 (140 > 75, 15/15), L-citrulline (176 > 70, 15/20), L-citrulline 13C5 D4 (181 > 75, 15/20), L-proline (116 > 70, 15/15), L- proline 13C5 15N (122 > 75, 15/15), L-phenylalanine (166 > 120, 15/15), L-phenyl-D5-alanine-D3 (174 > 128, 15/15), ADMA (203 > 46, 20/15) and 0.1 µM ADMA-D7 (210 > 77, 20/15). Quantification of AAs was performed by comparison of the relative area response obtained for the analyte versus the internal standard and calculated against a calibration curve for each analyte using Target Lynx software (Waters, Massachusetts, USA). All samples had concentrations within the calibration curve range.

### 4.4. Analysis of BH_2_ and BH_4_ in Plasma

For the analysis of BH_2_, 150 µL of 95% ACN with 0.2% DTT containing 10 ng/mL of the BH_2_-D3 internal standard was added to 25 µL ice-cold plasma in a polypropylene 96-well plate on ice. Samples were mixed for 5 min and centrifuged for 10 min at 4000 rpm at 4 °C. Then, 150 µL of the sample extract was transferred to a new deep-well plate ready for analysis by LC-MS. For the analysis of BH_4_, 250 µL of 95% ACN with 0.2% DTT was added to 25 µL ice-cold plasma in glass vials in a 96-well plate on ice. Samples were mixed for 5 min and centrifuged for 10 min at 4000 rpm at 4 °C. Then, 200 µL of the sample extract was transferred to new glass vials; 10 µL 500 mM ammonium carbonate and 10 µL benzoyl chloride (BC) were added, and samples were mixed for 1 min and left on ice for 15 min to allow the formation of a stable BH_4_-BC derivative. The reaction mixture was evaporated to complete dryness at 80 °C for about 2 h under a stream of nitrogen gas until 100% of all BC had evaporated completely. Then, 200 µL internal standard solution (1 ng/mL BH_4_-BC-D5 in 95% ACN with 0.1% DTE, without any traces of free BC) was added to the dry BH_4_-BC in the glass vials and mixed for 5 min at room temperature. Subsequently, 150 µL of the sample extract was transferred to a new deep-well plate, ready for analysis by LC-MS. BH_4_-BC-D5 was custom-made from BH_4_ derivatized with benzoyl chloride-D5, following the protocol by Yuan et al. [45], with a novel modification to dry all extracts to complete dryness to avoid the issue with false results resulting from the chemical activity of residual BC on the derivatives of BH_4_. Chromatographic separation was achieved by hydrophilic interaction chromatography (HILIC) using a 10 cm × 2.1 mm BEH amide column (Waters) in a gradient run, where mobile phase A was 95% acetonitrile with 5 mM ammonium formate (pH3) and mobile phase B was 10 mM ammonium formate (pH3) in water. The following gradient was applied for BH_2_ and BH_4_-BC, respectively, analyzed in separate samples: 0% B for 0–1 min, then from 0–20% B from 1–6 min and 20% B kept to 7 min, followed by equilibration from 7–10 min at 0% B prior to the next sample injection. Then, a 30 µL sample extract was injected using a large-volume injector. The sample plates were kept at 10 °C, and the column was held at 30 °C with a flow rate of 0.4 mL/min. The analyte was quantified in positive electrospray mode using multiple reaction monitoring (MRM) with the following MRM transitions, cone voltages and collision energies: BH_2_ (240.1 > 196.2, 20/12), BH_2_-D3 (243.1 > 168.2, 20/20), BH_4_-BC (346.2 > 166, 20/20) and BH_4_-BC-D5 (351.2 > 166.2, 20/20). Quantification of biopterines was performed by comparison of the relative area response obtained for the analyte versus the internal standard and calculated against a calibration curve for each analyte using Target Lynx software (Waters). All samples had concentrations within the calibration curve range.

### 4.5. In Vivo Endothelial Function and Stiffness

For the in vivo measurements of peripheral vascular endothelial function, the peripheral arterial tonometry–reactive hyperemia index (PAT-RHI) was calculated using an EndoPAT 2000 device (Itamar Medical, Caesarea, Israel). This method has previously been reported to correlate with endothelial function in the coronary circulation [17] and to be predictive for CVD [18]. Measurements were performed in the morning between 6 and 10 am, after an overnight fast, in a temperature-controlled room. Participants were asked to abstain from tobacco products and not to take their blood pressure medication. The EndoPAT device measures blood volume changes from the participant’s index fingers. A sphygmomanometer cuff was placed on the upper arm of the non-dominant arm and was inflated to 60 mmHg above systolic BP or at least 200 mmHg to induce reactive hyperemia, while the opposite arm served as control. In patients with arteriovenous fistula, the cuff was placed on the opposite arm. The signal was recorded for three 5-minute periods of: baseline recording, interruption of blood flow and recording of reactive hyperemia response. Computerized software with a proprietary algorithm automatically calculated the RHI from the fold increase in the pulse wave amplitude, relative to baseline and AI@75. The reference limit for RHI was set at >1.67, as specified by the manufacturer.

### 4.6. Ex Vivo Vascular Function and Structure

For the measurement of ex vivo functional and structural properties on isolated small arteries (e.g., endothelial and vascular smooth muscle function, contractility and stiffness), the multi-myograph (model 610M, Danish Myo Technology A/S, Hinnerup, Denmark) technique was used. During transplantation, a piece of subcutaneous fat was removed at the incision site. Resistance arteries (diameter ≈ 150–300 µm) were dissected from the fat biopsy and mounted in an organ bath of a four-vessel multi-myograph system for the measurement of endothelial, vascular smooth muscle cell function and structure, as previously described [46]. Briefly, concentration–response curves to norepinephrine (1 nM to 3 µM) and endothelium-dependent vasodilator bradykinin (BK) (10 pM to 1 µM) were performed before and after incubation with NOS inhibitor L-NG-nitro arginine methyl ester (L-NAME) (100 µM) and cyclooxygenase (COX) inhibitor indomethacin (100 µM, 20 min). For endothelium-independent dilation, concentration–response curves to sodium nitroprusside (SNP) (1 nM to 100 µM) were obtained. At the end of each experiment, a stretching procedure in the presence of 1 mM SNP, 0.2 mM papaverine and 1 mM ethylene glycol-bis β-aminoethyl ether (EGTA) in Ca^2+^-free PSS was performed to obtain the passive-length relationship to see the stiffness.

### 4.7. Statistical Analysis

Comparisons of clinical and biochemical markers, as well as in vivo experiments and comparing CKD stages to control participants, were assessed using the non-parametric Mann–Whitney U-test or chi-square test. For ex vivo studies, the Mann–Whitney U-test was used for group-wise comparisons, i.e., CKD vs. control. For the comparison of before and after NOS/COX inhibition, the Wilcoxon signed-rank test was used. Correlation analysis were performed using the non-parametric Spearman rank correlation method. All continuous variables were expressed as the median (interquartile range). Statistical significance was set at the level of *p* < 0.05. Graph pad Prism 6.0 (GraphPad Software Inc., Boston, MA, USA) and STATISTICA 7.0 (TX, USA) software were used for statistical analyses.

## Figures and Tables

**Figure 1 ijms-24-05582-f001:**
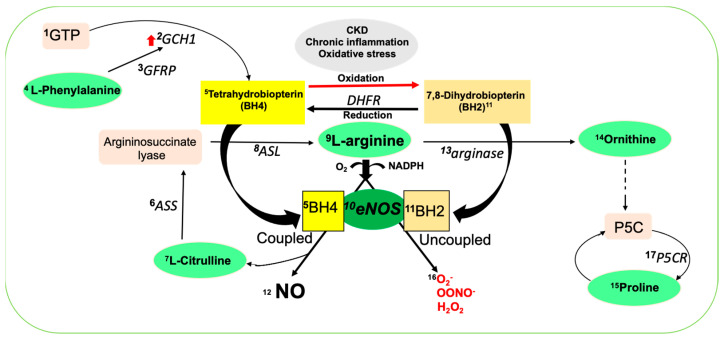
Schematic diagram of AA metabolism in endothelial cells and their role on NO generation. In physiological conditions, ^9^L-arginine produces ^12^NO and ^7^L-citrulline with the help of ^5^BH_4_ and coupled ^10^eNOS; in pathological conditions, i.e., in CKD, ^11^BH_2_ can uncouple eNOS and produce reactive free radicals^16^. Citrulline converts back to L-arginine with the help of ^6^ASS and ^8^ASL to keep the balance of NO. ^4^L-phenylalanine facilitates the production of BH_4_ with the help of ^1^GTP and different enzymes such as ^3^GFRP, ^2^GCH1. Excess L-arginine can also convert to ^14^ornithine and ^15^proline with the help of ^13^arginase and ^17^P5CR, respectively. Abbreviations: AA: amino acids; NO: nitric oxide; NOS: nitric oxide synthase; ASS: argininosuccinate synthase; ASL: argininosuccinate lyase; P5C:1-pyrroline-5-carboxylate; DHFR: dihydrofolate reductase; GTP: guanosine triphosphate; GCH1: GTP cyclohydrolase-1; GFRP: GTP cyclohydrolase 1 feedback regulatory protein; BH_2_: 7,8-dihydrobiopterin; BH_4_: tetrahydrobiopterin.

**Figure 2 ijms-24-05582-f002:**
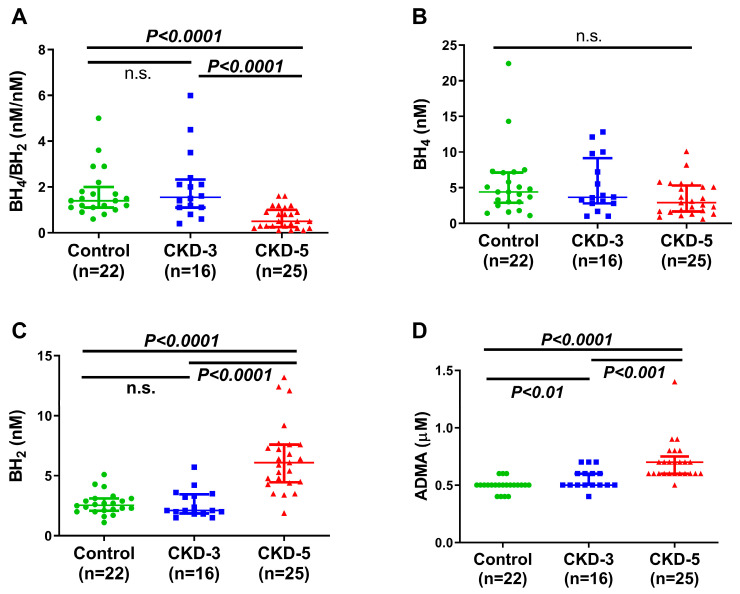
Plasma concentration of BH_4_, BH_2_ and ADMA. BH_4_/BH_2_ ratio (**A**), BH_4_ (**B**), BH_2_ (**C**) and ADMA (**D**) between control (n = 22), CKD-3 (n = 16) and CKD-5 (n = 25) groups, respectively. Data are expressed as median (interquartile range); significances were assessed by Mann–Whitney U-test. *p* < 0.0001 for BH_4_/BH_2_ and BH_2_ between control vs. CKD-5 and CKD-3 vs. CKD-5; for ADMA, *p* < 0.0001 between control vs. CKD-5, *p* < 0.01 between control vs. CKD-3 and *p* < 0.001 between CKD-3 vs. CKD-5. BH_4_, BH_2_ and ADMA were measured using chromatographic separation, and the BH_4_/BH_2_ ratio was calculated. BH_4_ = tetrahydrobiopterin; BH_2_ = dihydrobiopterin; ADMA = asymmetric dimethylarginine; CKD-3 = chronic kidney disease stage 3; CKD-5 = chronic kidney disease stage 5. n.s.: no significance.

**Figure 3 ijms-24-05582-f003:**
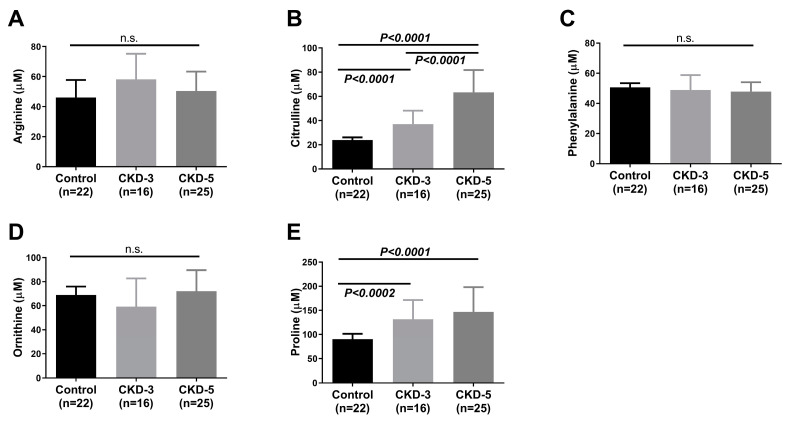
Plasma concentration of amino acids. Arginine (**A**), citrulline (**B**), phenylalanine (**C**), ornithine (**D**) and proline (**E**) between control (n = 22), CKD-3 (n = 16) and CKD-5 (n = 25) groups, respectively. Data are expressed as median (interquartile range); significances were assessed by Mann–Whitney U-test: for citrulline, *p* < 0.0001 between control vs. CKD-3 vs. CKD-5; for proline, *p* < 0.0001 between control vs. CKD-5 and *p* < 0.0002 between control vs. CKD-3. The concentration of arginine, citrulline, phenylalanine, ornithine and proline were measured using chromatographic separation. CKD-3 = chronic kidney disease stage 3; CKD-5 = chronic kidney disease stage 5. n.s.: no significance.

**Figure 4 ijms-24-05582-f004:**
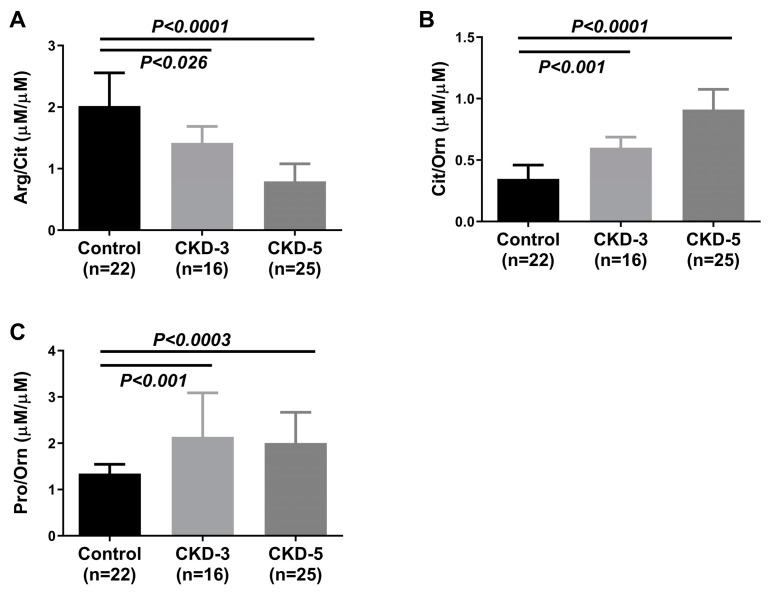
Plasma ratio of amino acids in control and CKD stages-3 and -5. (**A**) Arg/Cit, (**B**) Cit/Orn and (**C**) Pro/Orn ratios between control (n = 22), CKD-3 (n = 16) and CKD-5 (n = 25) groups, respectively. Data are expressed as median (interquartile range); significances were assessed by Mann–Whitney U-test. The significant differences were *p* < 0.0001 for Arg/Cit and Cit/Orn between control vs. CKD-5, *p* < 0.001 for Cit/Orn and Pro/Orn between control vs. CKD-3, *p* < 0.026 for Arg/Cit between control vs. CKD-3 and *p* < 0.0003 for Pro/Orn between control vs. CKD-5, respectively. The concentration of arginine, citrulline, ornithine and proline were measured using chromatographic separation, and the ratios of Arg/Cit, Cit/Orn and Pro/Orn were calculated from the measured concentration. Arg = arginine; Cit = citrulline; Pro = proline; Orn = ornithine; CKD-3 = chronic kidney disease stage 3; CKD-5 = chronic kidney disease stage 5.

**Figure 5 ijms-24-05582-f005:**
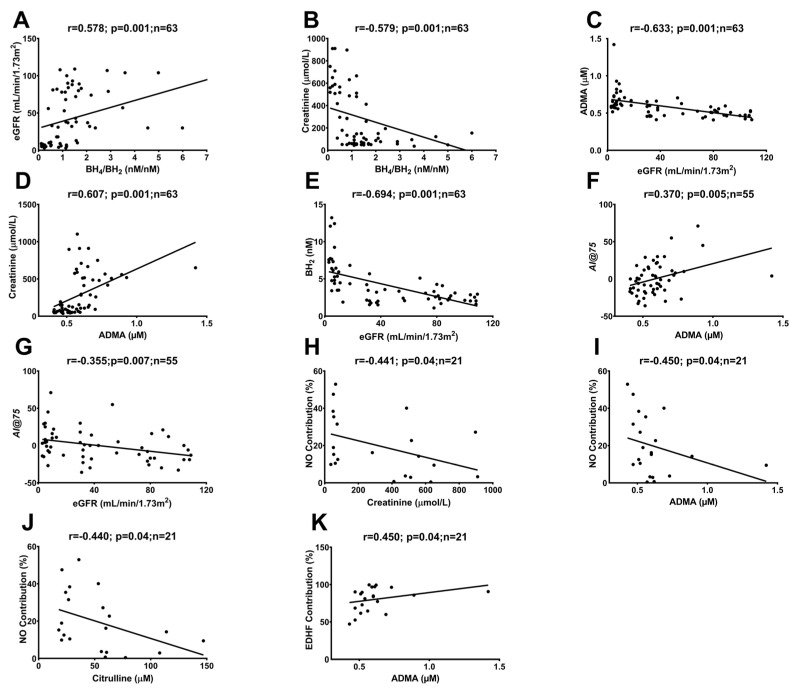
Correlations between the BH_4_/BH_2_ ratio with eGFR (r = 0.578; *p* = 0.001; n = 63) (**A**) and creatinine (r = −0.579; *p* = 0.001; n = 63) (**B**), ADMA with eGFR (r = −0.633; *p* = 0.001; n = 63) (**C**) and creatinine (r = 0.607; *p* = 0.001; n = 63) (**D**), BH_2_ with eGFR (r = −0.694; *p* = 0.001; n = 63) (**E**), AI@75 with ADMA (r = 0.370; *p* = 0.005; n = 55) (**F**) and eGFR (r = −0.355; *p* = 0.007; n = 55) (**G**), NO contribution with creatinine (r = −0.441; *p* = 0.04; n = 21) (**H**), ADMA (r = −0.450; *p* = 0.04; n = 21) (**I**) and citrulline (r = −0.440; *p* = 0.04; n = 21) (**J**), ADMA with EDHF contribution (r = 0.450; *p* = 0.04; n = 21) (**K**) in all participants. Spearman’s rank correlations were performed for the analysis. ADMA, citrulline and BH_4_/BH_2_ were measured using chromatographic separation; AI@75 was measured in vivo using EndoPAT and NO; EDHF contribution was measured ex vivo using the wire myograph technique. ADMA = asymmetric dimethylarginine; BH_4_ = tetrahydrobiopterin; BH_2_ = dihydrobiopterin; NO = nitric oxide; EDHF = endothelium-derived hyperpolarizing factor.

**Figure 6 ijms-24-05582-f006:**
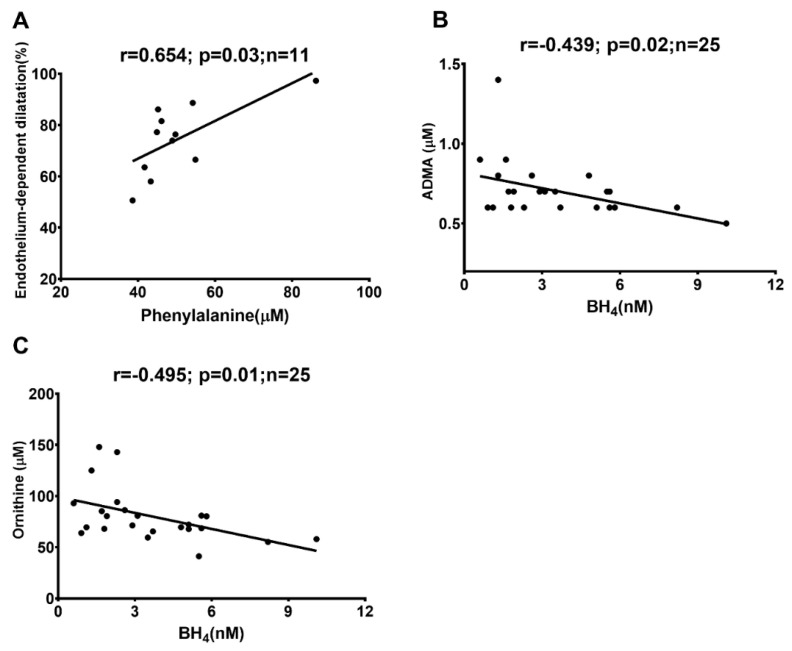
Spearman rank correlations between endothelium-mediated dilatation and phenylalanine (r = 0.654; *p* = 0.03; n = 11) (**A**), and BH_4_ with ADMA (r = −0.439; *p* = 0.02; n = 25) (**B**) and ornithine (r = −0.495; *p* = 0.01; n = 25) (**C**), respectively, in patients with CKD-5 only. ADMA, BH_4_ and ornithine were measured using chromatographic separation; the endothelium-dependent dilation was assessed using the ex vivo wire myograph technique. ADMA = asymmetric dimethylarginine; BH_4_ = tetrahydrobiopterin.

**Table 1 ijms-24-05582-t001:** Demographics and clinical characteristics of CKD-5, CKD-3 and non-CKD control participants.

	CKD-5 (n = 25)	CKD-3 (n = 16)	Control (n = 22)	*p*-Values	
				CKD-5 vs. Control	CKD-3 vs. Control
Demography					
Age (years)	48 (36–54)	44 (31–61)	42 (36–55)	0.99	0.87
Males (%)	18 (72%)	12 (75%)	6 (27%)	0.001	0.001
Body mass index (kg/m^2^)	25 (23–28)	24 (21–28)	26 (21–28)	0.95	0.54
eGFR(mL/min/1.73 m^2^)	7 (5–9)	35 (31–42)	100 (85–108)	0.001	0.001
SBP (mmHg)	146 (129–165)	137 (125–144)	120 (109–127)	0.001	0.02
DBP (mmHg)	93 (86–104)	78 (70–83)	77 (69–82)	0.001	0.16
Metabolic markers					
Creatinine (µM/L)	706 (573–903)	166 (141–211)	73 (67–81)	0.001	0.001
Total cholesterol (mM/L)	4.4 (3.4–5.3)	4.8 (4.5–6.1)	4.7 (4.4–5.2)	0.36	0.68
HDL cholesterol (mM/L)	1.2 (1.0–1.7)	1.4 (1.2–1.8)	1.6 (1.2–1.9)	0.05	0.60
Triglycerides (mM/L)	1.5 (0.9–1.8)	1.3 (1.1–2.0)	0.92 (0.7–1.3)	0.005	0.01
HbA1c, IFCC units (mM/M)	36 (33–38)	36 (32–37)	34 (32–36)	0.39	0.84
Inflammation					
hsCRP (mg/L)	0.8 (0.4–1.8)	0.9 (0.4–2.0)	0.57 (0.3–2.1)	0.67	0.16
Albumin (g/L)	36 (32–39)	32 (26–37)	40 (36–41)	0.005	0.001
Calcification & bone markers					
Calcium (mM/L)	2.3 (2.1–2.4)	2.2 (2.1–2.3)	2.3 (2.2–2.4)	0.45	0.64
Phosphate (mM/L)	1.8 (1.4–2.3)	1.2 (1.1–1.3)	0.9 (0.9–1.1)	0.001	0.01

All continuous data are given as median (interquartile range). Abbreviations: SBP = systolic blood pressure; DBP = diastolic blood pressure; eGFR: estimated glomerular filtration rate; HDL = high-density lipoprotein; HbA1c = glycated hemoglobin A1c; hsCRP = high-sensitivity C-reactive protein; CKD-3 = chronic kidney disease stage 3; CKD-5 = chronic kidney disease stage 5. Comparisons were assessed using the non-parametric Mann–Whitney U-test or chi-square test.

## Data Availability

The datasets generated and/or analyzed during the current study are available from the corresponding author upon reasonable request.

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
