# Peer review of "Associations of Biopterins and ADMA with Vascular Function in Peripheral Microcirculation from Patients with Chronic Kidney Disease"

_ijms, 2023, doi:10.3390/ijms24065582_

Round 1

Reviewer 1 Report

This study tests levels of plasma amino acids in CKD stages 3,5 and compares them with normal kidney function and their association with vascular and endothelial function through in vivo augmentation index. I have some queries. How was this population selected and sample size determined. What was the etiology of CKD. In methods it is mentioned that controls were selected with egfr > 100 but in table range starts from 85. How was GFR estimated and why was stage 4 not assessed. Range of GFR in CKD 3 is 28 to 45 in table 1. To be noted stage 3 is egfr 30 and above. Units for eGFR is given as kg but it should be ml/minute/m2. Authors have demonstrated signs of endothelial dysfunction and high levels of BH2, ADMA with a linear relation between the two. A note is required about novelty of these findings

Author Response

Reviewer 1:

This study tests levels of plasma amino acids in CKD stages 3,5 and compares them with normal kidney function and their association with vascular and endothelial function through in vivo augmentation index. I have some queries. How was this population selected and sample size determined. What was the etiology of CKD. In methods it is mentioned that controls were selected with egfr > 100 but in table range starts from 85. How was GFR estimated and why was stage 4 not assessed. Range of GFR in CKD 3 is 28 to 45 in table 1. To be noted stage 3 is egfr 30 and above. Units for eGFR is given as kg but it should be ml/minute/m2. Authors have demonstrated signs of endothelial dysfunction and high levels of BH2, ADMA with a linear relation between the two. A note is required about novelty of these findings

1. How was this population selected and sample size determined.

Our Comments: Thank you for the comment. This study was a cross-sectional study of an existing ongoing cohort of patients with CKD stage 5, and 3, while kidney donors served as non-CKD controls. For determining the sample size, before starting the project the initial decision was to include (n=20) participants in each group. The sample size of 20 in each group was considered appropriate for several reasons, first, the study used a qualitative research design, and the goal was to gather in-depth data from each participant. Thus, a selected sample size was deemed appropriate, as it allowed for a more comprehensive analysis of the data collected. Furthermore, recruiting more participants would have been challenging due to logistical and time constraints, particularly those with CKD-3. In addition, scientific literatures for similar type of study also supports our selection of sample size. A time frame was also decided from February 2019 to February 2021 for the collection of the samples, which indeed coincidence with some restriction of clinical studies related to COVID-19 situation. The specialist nephrologist together with research nurses at the Division of Renal Medicine, Karolinska University Hospital Sweden selected the participants for each group after investigating the clinical profile of the patients. All patients came to Renal Medicine Division during February 2019 to February 2021 were offered to participate in this study. To reduce selection bias, there was no exclusion criteria other than unwillingness to participate in the study were applied to the selection of the participants. The kidney donors were selected by as control. Children were not included in the study.  

2. What was the etiology of CKD?

Our Comments: The definition of the etiology of CKD was performed by specialist nephrologist at the Division of Renal Medicine, Karolinska University Hospital Sweden, considered several factors and underlying conditions that can contribute to kidney damage, such as diabetes, high blood pressure, and certain medications. In addition, they investigated various diagnostic tests used to assess kidney function and identify potential causes of CKD, such as blood and urine tests, imaging studies, and in some cases kidney biopsies. By considering all of this relevant information, our clinicians developed accurate understanding of the underlying causes of CKD for each patient. There were 1 and 2 participants with diabetic nephropathy in the CKD-5, and CKD-3 groups respectively.

2. In methods it is mentioned that controls were selected with egfr > 100 but in table range starts from 85. How was GFR estimated and why was stage 4 not assessed.

Our Comments: As mentioned earlier the kidney donors served as control in this study. Every kidney donor who has given informed consent during the period of sample collection were included in the control group. Although the eGFR of few control participant’s were <100 they showed no signs of kidney damage. We have now corrected the statement for control in the method section. Now it is written ‘‘controls were selected with eGFR> 85’’. The eGFR was calculated using age, sex and serum creatinine (μmol/L).

4. Range of GFR in CKD 3 is 28 to 45 in table 1. To be noted stage 3 is egfr 30 and above. Units for eGFR is given as kg but it should be ml/minute/m2.

Our Comments: Thank you for the comment and we do apologize for this inconsistency, we appreciate indeed the notice!  We have rechecked again the individual patient’s data related to calculation of eGFR, and indeed excluded one participant from the CKD-3 group as the eGFR of that participant was below 30 mL/min/1.73m2 (Please see the updated table). We have also included a sentence about this exclusion in the method section. However, this exclusion did not change the results, and the relevant figures with legends were remade in accordance. The unit of eGFR in the table was a typing mistake, we have now corrected the unit mL/min/1.73m2 in the revised version of the manuscript.

5. Authors have demonstrated signs of endothelial dysfunction and high levels of BH2, ADMA with a linear relation between the two. A note is required about novelty of these findings

Our Comments: The negative association between ADMA and endothelial function suggests that higher levels of ADMA may contribute to endothelial dysfunction in microcirculation and subsequent cardiovascular disease. This finding highlights the potential importance of reducing ADMA levels through lifestyle modifications or pharmacological interventions as a strategy for preventing or treating cardiovascular disease. The finding of a negative association between ADMA and endothelial function is not entirely novel, as previous studies have reported similar findings. However, majority of the studies to dates are either in animals or in bigger arteries that’s why it is still an important and relevant finding, that contributes to our understanding of the pathophysiology of cardiovascular disease and may inform future prevention and treatment strategies.

We did not report about any association between endothelial dysfunction and high levels of BH2, in the results.

Reviewer 2 Report

This review is so interesting and well-designed research.

#1  I think that it is better to examine the correlation using eGFR rather than Cre.

#2 NO synthesis is more injured in patients with diabetes versus non-diabetes. Are value of biopterins  significantly different between patients with and without diabetes?

Author Response

Reviewer 2:

This review is so interesting and well-designed research.

#1. I think that it is better to examine the correlation using eGFR rather than Cre.

Our Comments: Thank you for the suggestion. We have now performed correlation analysis with eGFR and updated the figure 5 in the revised version of the manuscript.  

#2. NO synthesis is more injured in patients with diabetes versus non-diabetes. Are value of biopterins significantly different between patients with and without diabetes?

Our Comments: Thank you very much for this valuable comment. Unfortunately, we were unable to perform the analysis between patients with and without diabetes as there were only 1 and 2 diabetes participants in the CKD-5 and CKD-3 groups respectively.

Reviewer 3 Report

In the manuscript: “Associations of biopterins and ADMA with vascular function 2 in peripheral microcirculation from patients with chronic kid-3 ney disease”. The authors examine the reduction of biopterin (BH4/BH2) ratio, founding an increase of BH2 plasma levels, asymmetric dimethylarginine (ADMA), and citrulline in patients with CKD-5 vs. CKD-3, which is related to the altered levels of amino acids and its metabolites in CKD patients. Moreover, authors test their relationship with endothelial and vascular smooth muscle function. An interesting topic that contributes to knowledge in the area, but certain issues must be corrected.

Major revisions

1.    In the abstract and introduction, an introduction and the gap of the manuscript must be mentioned; that is, the authors must mention the gap that the manuscript will fill within the current knowledge.

2.    In figure 1, the authors should place numbers that readers could follow when reading the figure caption and thus be able to read the figure easily.

Please check grammatical errors; for example, in figure 1 citrulline is not well written.

The figure must be placed after it is spoken of. In the manuscript, the figure first appears, and then the reference is made to it.

3.    The results should be improved by adding an explanation at the beginning of their description, mentioning why the assay in question was used and why the authors decided to perform that assay and not another.

4.    The results must have concluded. That is, authors must mention the conclusions of each result: for instance, these results together suggest… or we concluded that…

5.    In the figure captions, the number of experiments that were used for the corresponding statistical tests must be mentioned. Moreover, the authors must add the statical test that was used in the assays.

6.    Please add the number of samples (n=x), statistical test, statistical significance, and a brief description of the experiments performed in the experiments in the figure legends.

7. Authors must explain because they did not make the experiments ex vivo for the control. It is a big issue.

8.    The authors are also encouraged to measure invasion markers such as metalloproteases.

9.    Statements on lines 281-283 need a reference.

10. Please add a conclusion, not a summary.

Author Response

Reviewer 3:

Major revisions

1. In the abstract and introduction, an introduction and the gap of the manuscript must be mentioned; that is, the authors must mention the gap that the manuscript will fill within the current knowledge.

Our Comments: Thank you for this suggestion, we have now included a sentence both in abstract and introduction mentioning about the research gap and how our study can contribute to it in the revised version of the manuscript. Line 1-13 and 99-108

2. In figure 1, the authors should place numbers that readers could follow when reading the figure caption and thus be able to read the figure easily.

Please check grammatical errors; for example, in figure 1 citrulline is not well written.

The figure must be placed after it is spoken of. In the manuscript, the figure first appears, and then the reference is made to it.

Our Comments: We have now placed numbers in figure 1 and checked grammatical errors. The figure is moved to the place as appropriate in the revised version of the manuscript. 

3. The results should be improved by adding an explanation at the beginning of their description, mentioning why the assay in question was used and why the authors decided to perform that assay and not another.

Our Comments: Thank you for this comment, we have now added an explanation at the beginning of each result in the revised version of the manuscript.

4. The results must have concluded. That is, authors must mention the conclusions of each result: for instance, these results together suggest… or we concluded that…

Our Comments: We have already included a conclusion/suggestion for each finding in the discussion part. However, now we included conclusion/suggestion for some findings in the results where appropriate in the revised version of the manuscript.

5. In the figure captions, the number of experiments that were used for the corresponding statistical tests must be mentioned. Moreover, the authors must add the statical test that was used in the assays.

Our Comments: If we understand correctly comment 5 and 6 looks similar to us. We have addressed those concerns in the revised version of the manuscript.

6. Please add the number of samples (n=x), statistical test, statistical significance, and a brief description of the experiments performed in the experiments in the figure legends.

Our Comments: Thank you for the comments. We have now included number of samples, statistical test, statistical significance and brief description of the experiments performed in the figure legends in the revised version of the manuscript.

7. Authors must explain because they did not make the experiments ex vivo for the control. It is a big issue.

Our Comments: There must be a slight misunderstanding, we have performed ex vivo experiments for both the control and CKD-5 groups. However due to unavailability of the biopsies we did not perform ex vivo experiment for the CKD-3 group. Line 213-214

8. The authors are also encouraged to measure invasion markers such as metalloproteases.

Our Comments: Thank you very much and really appreciated for the suggestion however, at this moment we are unable to run metalloproteases experiment. In future, we will consider this in different project. 

9. Statements on lines 281-283 need a reference.

Our Comments: We have now included reference for the statements on lines 281-283 in the revised version of the manuscript. Line 340-343

10Please add a conclusion, not a summary.

Our Comments: We have now added a concluding sentence in the revised version of the manuscript. Line 343-344

Round 2

Reviewer 3 Report

The authors have made the revisions that concerned me. Thanks!